# Lymphatic filariasis endgame strategies: Using GEOFIL to model mass drug administration and targeted surveillance and treatment strategies in American Samoa

**Callum Shaw**[1]*, **Angus McLure**[1], **Patricia M. Graves**[2], **Colleen L. Lau**[3], **Kathryn Glass**[1]

**1** National Centre for Epidemiology and Population Health, Australian National University, Canberra, ACT, Australia, **2** College of Public Health, Medical and Veterinary Sciences, James Cook University, Cairns, Queensland, Australia, **3** School of Public Health, Faculty of Medicine, The University of Queensland, Brisbane, Queensland, Australia

* callum.shaw@anu.edu.au

**Data Availability Statement:** The GEOFIL code and all model output used in this study are available on

## Abstract

American Samoa underwent seven rounds of mass drug administration (MDA) for lymphatic filariasis (LF) from 2000-2006, but subsequent surveys found evidence of ongoing transmission. American Samoa has since undergone further rounds of MDA in 2018, 2019, and 2021; however, recent surveys indicate that transmission is still ongoing. GEOFIL, a spatially-explicit agent-based LF model, was used to compare the effectiveness of territory-wide triple-drug MDA (3D-MDA) with targeted surveillance and treatment strategies. Both approaches relied on treatment with ivermectin, diethylcarbamazine, and albendazole. We simulated three levels of whole population coverage for 3D-MDA: 65%, 73%, and 85%, while the targeted strategies relied on surveillance in schools, workplaces, and households, followed by targeted treatment. In the household-based strategies, we simulated 1-5 teams travelling village-to-village and offering antigen (Ag) testing to randomly selected households in each village. If an Ag-positive person was identified, treatment was offered to members of all households within 100m-1km of the positive case. All simulated interventions were finished by 2027 and their effectiveness was judged by their 'control probability'—the proportion of simulations in which microfilariae prevalence decreased between 2030 and 2035. Without future intervention, we predict Ag prevalence will rebound. With 3D-MDA, a 90% control probability required an estimated ≥ 4 further rounds with 65% coverage, ≥ 3 rounds with 73% coverage, or ≥ 2 rounds with 85% coverage. While household-based strategies were substantially more testing-intensive than 3D-MDA, they could offer comparable control probabilities with substantially fewer treatments; e.g. three teams aiming to test 50% of households and offering treatment to a 500m radius had approximately the same control probability as three rounds of 73% 3D-MDA, but used < 40% the number of treatments. School- and workplace-based interventions proved ineffective. Regardless of strategy, reducing Ag prevalence below the 1% target threshold recommended by the World Health

GitHub (https://github.com/AngusMcLure/GEOFIL/tree/Targeted).

**Funding:** CS was supported by an Australian Government Research Training Program (AGTRP) scholarship. AM was supported by an Australian Research Council Discovery Project Grant (DP180100246). CLL was supported by an Australian National Health and Medical Research Council Fellowship (1109035). PG and KG received no specific funding for this work. The funders had no role in study design, data collection and analysis, decision to publish, or preparation of the manuscript.

**Competing interests:** The authors have declared that no competing interests exist.

Organization was a poor indicator of the interruption of LF transmission, highlighting the need to review blanket elimination targets.

## Author summary

Lymphatic filariasis (LF) is a parasitic disease caused by infection with filarial worms and is currently endemic in 72 countries. Mass drug administration (MDA) is used to interrupt LF transmission, in order to reduce LF prevalence below a target threshold set by the World Health Organization. When prevalence is below said threshold, continued LF transmission is believed to be unsustainable. American Samoa implemented seven rounds of MDA from 2000–2006 and, more recently, a further three rounds of MDA in 2018, 2019, and 2021. Despite these recent interventions, American Samoa has not yet reached elimination targets. In this study we use GEOFIL, a LF transmission model specific to American Samoa, to investigate the most efficient interventions for LF elimination—whether to continue with MDA or instead try targeted surveillance and treatment strategies. We find that, without further intervention, LF prevalence will rise, but further rounds of MDA have the potential to eliminate LF. Household-focused targeted surveillance and treatment interventions also have the ability to eliminate LF using far fewer treatments than MDA; however, they require testing a large number of people. Furthermore, we found that reducing antigen prevalence below threshold targets does not necessarily guarantee successful long-term elimination.

## Introduction

Lymphatic filariasis (LF) is a vector-borne parasitic disease affecting millions of people globally. LF is caused by infection with worms of any of three filarial parasites: *Wuchereria bancrofti*, *Brugia malayi*, or *B. timori*. The parasites are transmitted by several genera of mosquito. After infection, the larval worms mature in the lymphatic system where they become reproductively active, producing millions of microfilariae (mf) [1]. The mf circulate in the body and can infect mosquitoes if subsequent biting events occur. LF is currently endemic in 72 countries [2] where an estimated 51 million people were infected in 2018 alone [3]. Of those infected, many suffer a range of disabilities, with 36 million people suffering from severe lymphoedema [4].

The Global Programme to Eliminate LF (GPELF), was founded in 2000 with the aim of eliminating LF as a public health problem. The program had two primary goals: i) interrupt transmission with mass drug administration (MDA) and ii) alleviate the suffering of infected people. From the inception of GPELF, MDA has consisted of annual rounds of treatment with diethylcarbamazine and albendazole (2D-MDA) or with ivermectin and albendazole (IA) in areas with onchocerciasis [5]. Since 2018, triple-drug MDA with ivermectin, diethylcarbamazine, and albendazole (3D-MDA) has been implemented, as it can improve MDA effectiveness [6]. Post-MDA, several years of surveillance are required, which often involves transmission assessment surveys (TAS) that focus on detection of antigen-positive children aged between six and seven years. Countries pass TAS if the prevalence of infection is below target thresholds set by the World Health Organization (WHO). In *W. bancrofti* regions where *Anopheles* and/or *Culex* are the principal vectors, the target threshold is $< 2\%$ antigen prevalence, whereas in areas where *Aedes* is the primary vector, the target threshold is $< 1\%$ antigen prevalence [7].

Below the target threshold, continued transmission is believed to be unsustainable; however, previous modelling suggests that critical prevalence thresholds are highly dependent on local conditions [8–11].

Historically, the Pacific had among the highest prevalence of LF globally [12]. In 1999, the WHO launched the Pacific Programme to Eliminate LF. At the time, LF was endemic in 16 of the 22 Pacific nations [13]. Through the use of several rounds of MDA in the early to mid 2000s, Vanuatu, Niue, the Cook Islands, Tonga and the Marshall Islands were declared to have eliminated LF as a public health problem (EPHP), while several other Pacific nations were assessed as being close to achieving EPHP [14]. By 2019, eight of the sixteen endemic countries had received WHO validation that EPHP had been achieved [15]; however, despite this progress, several challenges remain for LF elimination in the Pacific.

American Samoa underwent seven rounds of 2D-MDA from 2000 to 2006 and passed TAS-1 in 2010 and TAS-2 in 2015 [16], but failed TAS-3 in 2016 [17]. This result, coupled with other community surveys, indicated that LF was still endemic [18, 19]. American Samoa subsequently implemented two rounds of 3D-MDA in 2018 and 2019; however, a 2019 community survey found transmission was still ongoing [20]. There was a further round of 3D-MDA in 2021, but it is unclear if this additional round will be enough to ensure long-term LF elimination [21].

There are a number of theorised causes for ongoing LF transmission in American Samoa. Firstly, previous MDA rounds may have suffered from low coverage [22]. Secondly, in American Samoa LF is transmitted by several vectors including the highly efficient *Aedes polynesiensis* that are predominantly active during the day and *Aedes samoanus* at night. In low prevalence settings, as currently in American Samoa, *Aedes* mosquitoes have been found to be particularly effective vectors [23–26]. Furthermore *W. bancrofti* larvae development has been found to be more efficient in *Aedes* mosquitoes than in other mosquito genera [27]. Additionally, while American Samoa as a whole might be below the target antigen prevalence threshold following MDA, there could be individual villages or localised areas above the target threshold. These localised 'hotspots' have the potential to drive continuing transmission of LF, due to the effectiveness of the *Aedes* vector. LF transmission hotspots have been found in many countries [16, 28–34] and range in distance from 10 metres [32] to one kilometre [30]. Finally, previous studies have found that standard target thresholds may not be suitable for all settings [29, 35, 36], as there have been previous resurgences of LF after antigen prevalence was reduced below the recommended thresholds [12].

In order to eliminate LF, American Samoa could undergo several more rounds of territory-wide MDA; however, a recent study found surveillance of certain sub-populations in American Samoa is highly effective in identifying residual infections [37]. Therefore, future LF strategies may benefit from both targeted surveillance, which can be used to inform targeted treatment, and accounting for the unique factors driving LF transmission in American Samoa. These targeted strategies could be more effective than traditional MDA in the low prevalence setting of American Samoa and may help avoid further burdening health systems [38].

In this study we use GEOFIL, a spatially explicit agent-based model [39]. Unlike other LF models, GEOFIL includes geographic information on household, workplace and school locations, daily commuting networks [40], and a spatially heterogeneous risk of infection. These factors allow GEOFIL to consider potential endgame surveillance and treatment frameworks in American Samoa that address geographic heterogeneity and hotspots. We developed three different targeted strategies that relied on surveillance in schools, workplaces, and households, followed by targeted treatment. We then compared further rounds of 3D-MDA with the targeted strategies to determine the most efficient route to long-term LF control in American Samoa.

## Methods

GEOFIL is a spatially explicit agent-based modelling framework, designed to model LF transmission dynamics in American Samoa. To capture American Samoan population behaviour, GEOFIL uses a synthetic population model that allows for births, deaths, couple formation and separation, moving within American Samoa, immigration, and emigration [39, 41]. In GEOFIL people are assigned households, workplaces, and schools; the locations of these buildings correspond to known locations of buildings in American Samoa. During working hours people are at their day-time location (workplace for employed, school for students, or household for all other people) and during off-work hours people are at their night-time location (household for all people). The local geographic area around these buildings is where transmission can occur.

To effectively model LF transmission in the human-agent population, GEOFIL is run on a daily time step and explicitly models the human-agent while implicitly modelling the mosquito-vector. Every time-step the model accounts for synthetic population changes, transmission, and the worm life-cycle within the host (acquisition, maturation, fecundity, and mortality). As transmission can occur at either a person's day-time location or night-time location, transmission during working hours and off-work hours is modelled separately. During each period, people may have zero, one, or multiple transmission events, where a transmission event is an infectious bite that transmits either one or two third stage (L3) larvae that survive to maturity. The number of transmission events an individual is exposed to in each period is a function of the local prevalence of infectious persons in neighbouring buildings (inversely scaled by distance), the time dependent transmission rate (day-time or night-time), and the age-dependent relative biting rate. These factors allow for a spatially and temporally heterogeneous risk of infection. Further details on the synthetic population, transmission dynamics, and model initialisation are given in S1–S3 Texts.

### Fitting the GEOFIL model

As appropriately modelling the clustering of LF cases was key to assessing the effectiveness of targeted strategies, we refitted the previous GEOFIL model [21] to reproduce the degree of clustering at the household and village level observed in a previous 2016 community survey [37]. As we only had intra-cluster correlation (ICC) values for microfilariae (mf) positivity from the 2016 survey, the model was initialised in 2010 with prevalence data from a 2010 community survey [35], but with village and household level mf ICC values from the 2016 survey.

GEOFIL was fitted using approximate Bayesian computation (ABC), to results from two LF community surveys in American Samoa. The surveys were conducted in 2014 [17] and in 2016 [37], and the model was fitted to the number of antigen and mf positive persons each survey found. The 2016 survey was also used to fit the levels of mf clustering in the model. A more detailed explanation of the ABC-fitting procedure used in this study is given in S4 Text.

### Mass drug administration scenarios

The model was configured to explore the effectiveness of different 3D-MDA coverages. In this study, coverage is calculated over the whole simulated population, but only the eligible population received treatment. In accordance with WHO recommendations children below two years of age and pregnant women were excluded from MDA, children aged two to four years were offered 2D-MDA (diethylcarbamazine and albendazole), while those aged five years and older were offered 3D-MDA [42]. Three different coverage levels were tested: the WHO recommended minimum of 65%, the previously reported coverage of 73% in a 2018 survey [43],

and a high coverage of 85%. For each coverage level, we modelled one to five annual rounds starting in 2023.

## Workplace-based surveillance and treatment strategies

Workplaces can provide an effective setting to gauge LF prevalence in American Samoa [44]. As such, GEOFIL was extended to include workplace-based surveillance and targeted treatment strategies. In GEOFIL, workplace commutes are highly heterogeneous and cover the whole of the main island of Tutuila [39], which allows workplaces to have workers from households all over the island. The workplace-based strategies relied on annual mass testing of workers. Two distinct classes of workplace were analysed: large workplaces (50+ employees) and small & large workplaces (5+ employees); workplaces with less than five employees were not considered for logistical reasons. For both classes, the workplaces were tested annually for five years and antigen-positive workers were treated. The two classes of workplaces were then simulated with a second strategy, where if an Ag-positive worker was found, the worker and members of their household would receive treatment. Both workplace strategies assumed 73% of workers would accept testing, and if a worker had accepted testing it was assumed that the worker and all members of their household would accept treatment. Like 3D-MDA, persons older than five years were treated with ivermectin, diethylcarbamazine, and albendazole (3D-treatment), while those between two and four years were treated with diethylcarbamazine and albendazole (2D-treatment).

## School-based surveillance and treatment strategies

Three distinct school-based surveillance strategies were investigated. The strategies involved testing different age brackets of students and if a child was Ag-positive, the child and their household would receive treatment. The first age bracket tested was TAS-aged children (6–7 years), the second was elementary school aged children (6–13 years), and thirdly, all school aged children (6–17 years) were tested. GEOFIL assumes all children aged 6–17 years attend school, apart from a small proportion of secondary school aged children who are instead employed [39]. Therefore, the school-based strategy would capture most children. Importantly, unlike workplaces, the GEOFIL model assumes that school enrolment is localised, with children attending the closest school to their household [39].

The simulations assumed a 73% probability that a child would accept testing, and if a child tested Ag-positive it was assumed that the child and all members of their household would accept treatment. Persons older than five years were treated with 3D-treatment, while those between the ages of two and four years were treated with 2D-treatment.

## Household-based surveillance and treatment strategies

We configured a household-based strategy, in which a number of teams would go from village-to-village offering testing to a proportion of households in each village (household test aim; HTA), with the goal of identifying and treating any residual infections and hot-spots. All members of a household were assumed to make the same decision to accept or reject testing, though testing was not offered to persons under six years old. If members of a household accepted testing it was assumed they would all accept treatment; similarly if members of a household rejected testing it was assumed they would also all reject treatment. As with previous strategies, the probability of a household accepting testing or treatment was 73%.

If an antigen-positive person was found, every member of their household was treated and members of all households within a certain radius (100 metres to 1 kilometre) around their positive-household were offered treatment. Similar to previous targeted strategies, persons

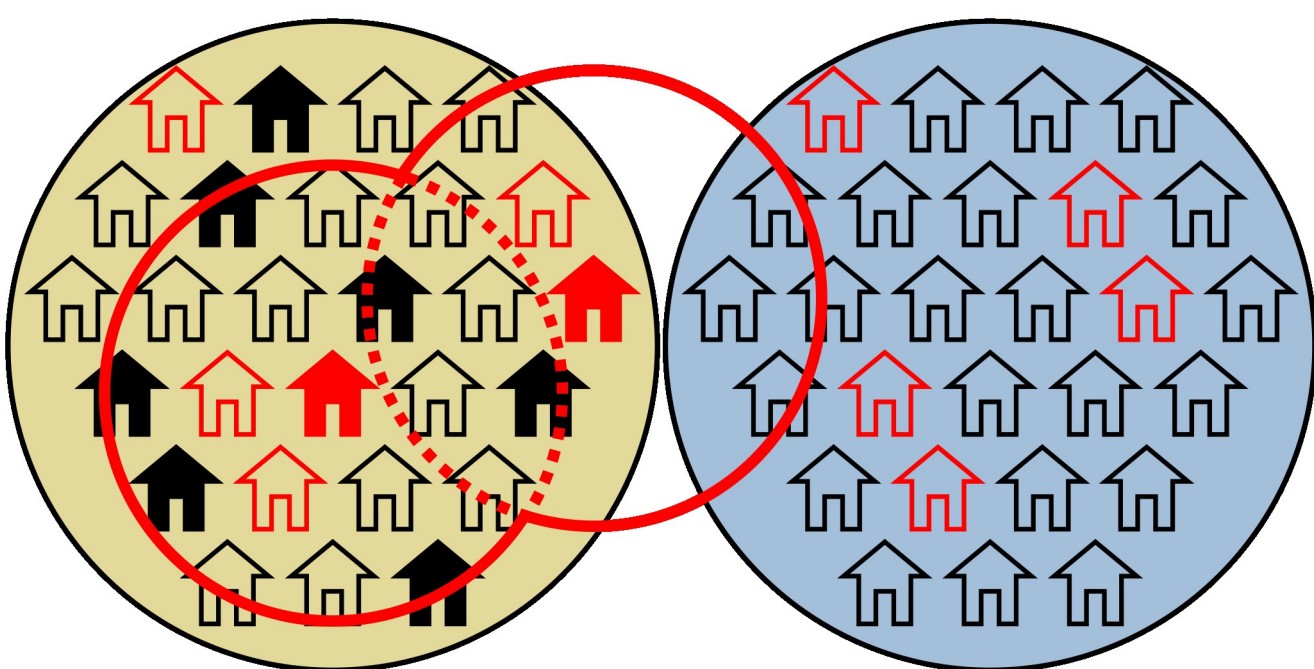

**Fig 1. Illustration of household-based strategy.** Figure depicts households in two neighbouring villages; household with antigen-positive members are outlined in red, whereas those free from LF have a black outline. The team is currently in the village on the left and tests a proportion of all households. Households that accept testing are shown with solid fill– black if there is no antigen positive members or red if there is an antigen positive household member. If the team identifies a household with an antigen-positive member, the team will offer treatment to members of all households within a given radius around the positive household, except to members of households that recently tested negative or had refused a test. This treatment radius is marked in red. As shown in the diagram, teams will offer treatment to members of households in neighbouring villages that are within the treatment radius of a LF-positive household.

older than five years were treated with 3D-treatment, while those between two and four years were treated with 2D-treatment. Households that had recently tested negative or refused testing would not receive treatment. Once a household was offered testing and/or treatment, the team would not offer testing or treatment to that household for a minimum of 180 days. A diagram illustrating the household-based procedure is shown in Fig 1.

The household-based strategy was simulated for five years (2023–2027) with one, three, or five teams using a 100m or 500m treatment radius. The one-team and three-team schemes were also simulated with a 1km treatment radius. This 1km radius was omitted for the five-team strategy due to the small area of American Samoa and the frequent overlaps in treatment it caused. All strategies were run with a household test aim of 25% and 50%. Additional household-based strategy model parameters are given in Table 1.

As the household-based strategy, by design, is testing-intensive, a second less testing-intensive method was modelled, where testing was only offered to a single adult in each household, instead of the entire household. Again, if an Ag-positive person was found, treatment was offered to their entire household and to all households within a certain radius.

## Modelled scenarios

All modelled strategies began in 2010 and included three initial rounds of MDA in 2018, 2019, and 2021. These initial MDA rounds were set to have 73% coverage, the surveyed coverage from the 2018 MDA round [43]. All additional strategies began in 2023 and each was run 100 times. A summary of all strategies is given in Fig 2. To gauge the effectiveness of each

**Table 1. Parameters used for the household-based intervention approach.**

| Variable | Value(s) | Description |
|---|---|---|
| Coverage | 73% | Probability a member of a household will accept a test or treatment. If the first member accepts or rejects, all other household members will have the same response |
| Household test aim (HTA) | 25%, 50% | Proportion of households that are are offered testing in each village |
| Household testing intensity | 1 adult, whole household | The number of household members tested in each household |
| Radius | 100m, 500m, 1km | The treatment radius around positive households |
| Minimum households | 10 houses | The minimum number of households offered testing in a village. If fewer than 10 households, all households will be offered testing |
| Households per day | 15 houses | Average number of households tested or treated per day |
| Minimum time | 14 days | The minimum time a team spends in each village. Purposely high to account for weekends and holidays |
| Days until Return | 180 days | The time until a household can be tested or treated again |

approach, a baseline simulation was conducted that contained no additional intervention after the 2021 MDA round. All strategies were completed by 2027 and their effectiveness was judged using a single metric, *control probability*, defined as the proportion of simulations in which mf prevalence decreased between 2030 and 2035 (the final simulation year).

## Results

### Fitting the GEOFIL model

To achieve the desired household and village mf intra-cluster correlations in GEOFIL, we found there had to be, on average, more transmission during off-work hours than during working hours. This unequal distribution in the timing of infection shifted the location where transmission was dominant, and caused an increase in transmission at the household and a decrease in transmission at schools and workplaces compared to previous versions of GEOFIL [21]. Further details on the results of the ABC fitting are given in S1 Fig.

### No further intervention

The baseline run, which only simulated 3D-MDA rounds in 2018, 2019, and 2021, each with 73% coverage, predicted that while territory-wide antigen prevalence would likely fall below 1% from 2024 to 2033, it would rebound and rise to 1.24%, [95% Prediction Interval (PI) 1.11–1.37] by 2035 if no further interventions were implemented (Fig 3, baseline curve).

### School- & workplace-based surveillance & treatment

School-based and workplace-based surveillance and treatment strategies were not able to control LF after five years of implementation; the control probabilities for these strategies were negligible. In all simulations, territory-wide antigen prevalence was estimated to fall below the WHO threshold of 1%; however, in all simulated strategies prevalence began to increase in the years proceeding treatment (Fig 3). Of the school-based interventions, the surveillance and treatment of TAS-aged children (6–7 years) was the least effective strategy with only a slight benefit over the baseline, with a predicted 2035 territory-wide antigen prevalence of 1.06%, [95% PI 0.97–1.20]. The testing and treatment of all elementary-school aged children was slightly more effective, with a predicted 2035 territory-wide antigen prevalence of 0.85%, [95%

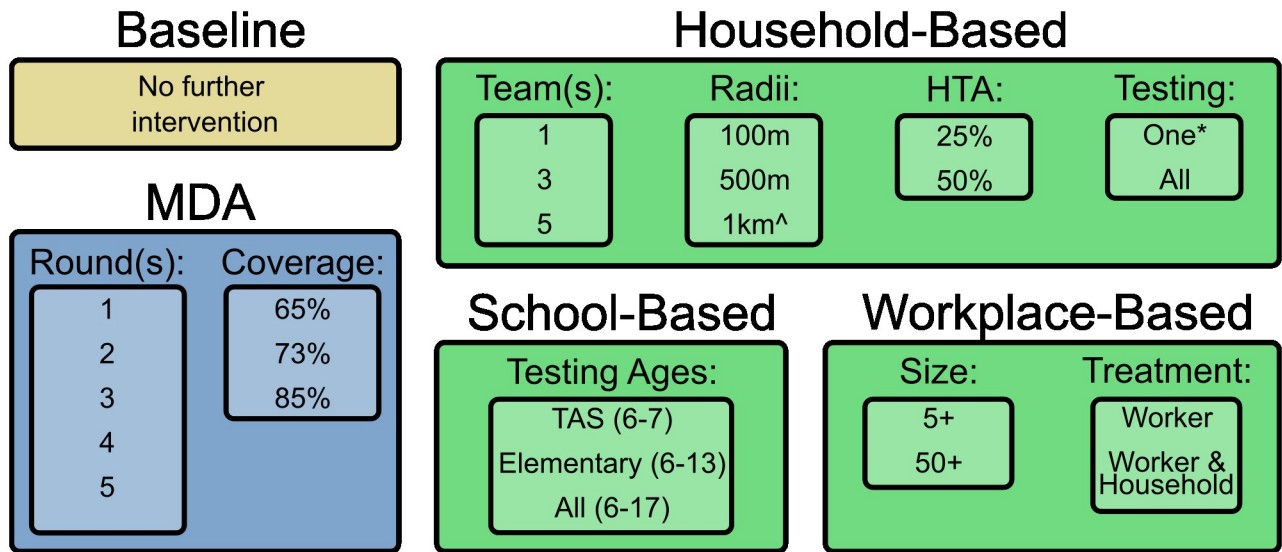

**Fig 2. Summary of all simulated surveillance strategies.** ^ The five-team household-based strategy was not simulated with the 1km treatment radius. *The household-based strategy where only a single member of each household was offered testing was not simulated with one-team.

PI 0.74–0.95]. Testing of all school aged children was the most effective school-based strategy and had a predicted 2035 territory-wide antigen prevalence of 0.72%, [95% PI 0.64–0.81].

Similar to the school-based interventions, workplace-based strategies were able to suppress an increase in antigen prevalence during treatment years; however, territory-wide antigen prevalence increased after treatment ended in 2027 (Fig 3). Of the workplace-based strategies,

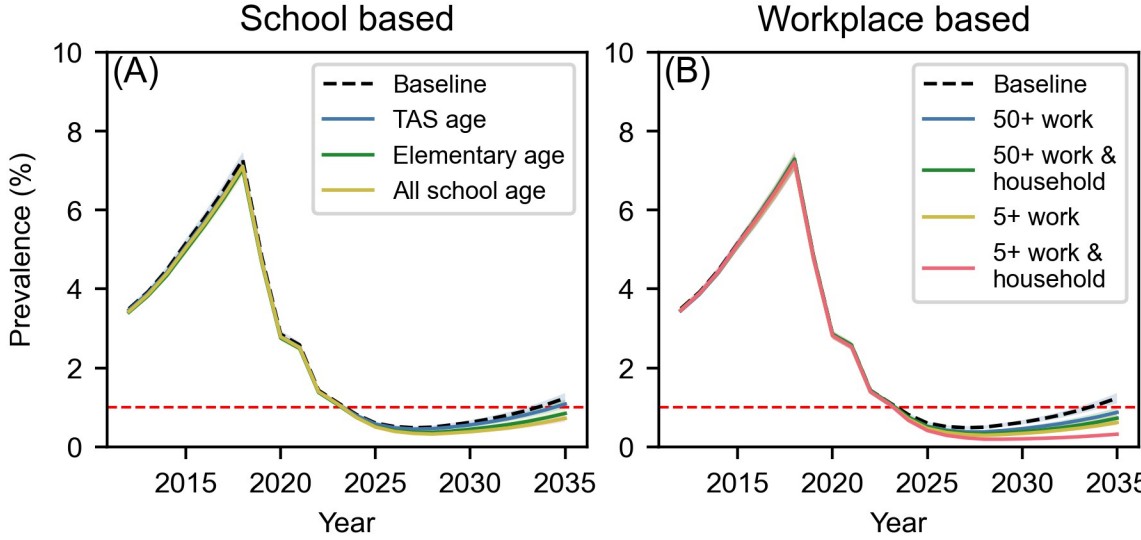

**Fig 3. Annual territory-wide antigen prevalence for school-based and workplace-based interventions.** Territory-wide antigen prevalence in the synthetic American Samoan population following three rounds of 3D-MDA in 2018, 2019, and 2021, followed by five rounds of targeted strategies. (A) school-based intervention that tests three different age groups of children (TAS-aged children (ages 6–7 years), elementary school aged children (ages 6–13 years), and all school aged children (ages 6–17 years)) and treats antigen-positive children and members of their household. (B) workplace-based intervention that tests workplaces with 50+ workers or 5 + works and either treats antigen-positive workers or the antigen-positive worker and members of their household. 1% antigen prevalence is shown by the red horizontal dashed line in both plots.

**Table 2. The effectiveness of additional rounds of annual 3D-MDA starting in 2023 for three different levels of treatment coverage.** A strategy's effectiveness is judged by its control probability, the proportion of simulations in which mf prevalence decreased between 2030 and 2035. Treatments are the average number of treatments administered over the duration of MDA and do not include treatment numbers from the prior MDA rounds in 2018, 2019, and 2021. The uncertainty in the control probabilities is quantified with a 95% credible interval.

| Rounds | 65% Coverage | | 73% Coverage | | 85% Coverage | |
|---|---|---|---|---|---|---|
| | Control probability (%) | Treatments | Control probability (%) | Treatments | Control probability (%) | Treatments |
| 1 | 11 [5.6–17.6] | 30,992 | 23 [15.3–31.6] | 34,911 | 54 [44.3–63.6] | 40,592 |
| 2 | 48 [38.4–57.7] | 61,653 | 59 [49.3–68.4] | 69,300 | 90 [83.6–95.2] | 80,691 |
| 3 | 71 [61.9–79.5] | 92,004 | 93 [87.4–97.2] | 103,331 | 98 [94.5–99.8] | 120,288 |
| 4 | 94 [88.7–97.9] | 121,782 | 97 [93.0–99.5] | 136,800 | 100 [98.1–100] | 159,336 |
| 5 | 100 [98.1–100] | 151,366 | 100 [98.1–100] | 169,982 | 100 [98.1–100] | 197,933 |

the most effective intervention was testing all workplaces with more than 5 workers and treating the worker and their household members. By 2028, territory-wide antigen prevalence under this strategy was predicted to be 0.19%, [95% PI 0.17–0.21], however, by 2035, the antigen prevalence was predicted to increase to 0.32%, [95% PI 0.28–0.36].

## Mass drug administration scenarios

MDA simulations found American Samoa required several more annual rounds of 3D-MDA to adequately control LF (Table 2). To achieve a $\geq$ 50% control probability required, on average, an estimated: three rounds at 65% coverage, two at 73% coverage, or one at 85% coverage. A $\geq$ 90% control probability required an estimated four rounds of 65% coverage, three rounds of 73% coverage, and two rounds of 85% coverage.

MDA with higher levels of coverage required fewer rounds and fewer total treatments to achieve a given control probability. However, above 75% control probability there was a treatment saturation, where additional rounds of MDA had a diminishing effect on control probability.

## Household-based surveillance and treatment strategies

With the correct scale of intervention, the household-based strategy could offer a reasonable control probability. Control probability improved with increases in the number of teams, household test aim (HTA), and treatment radii (Table 3). One-team strategies did not offer viable long-term LF control; although they were the least intensive in terms of tests and treatments, the estimated control probability was always < 40%. The three-team strategies could

**Table 3. The effectiveness of the household-based strategy, with different number of teams, treatment radii, and household test aims (HTA).** A strategy's effectiveness is judged by its control probability, the proportion of simulations in which mf prevalence decreased between 2030 and 2035. Treatments are the average number of treatments administered over the duration of the strategy and do not include treatment numbers from the prior MDA rounds in 2018, 2019, and 2021. The uncertainty in the control probabilities is quantified with a 95% credible interval.

| Teams | HTA | 100m Radius | | | 500m Radius | | | 1km Radius | | |
|---|---|---|---|---|---|---|---|---|---|---|
| | | Control probability (%) | Treatments | Tests | Control probability (%) | Treatments | Tests | Control probability (%) | Treatments | Tests |
| 1 | 25% | 9 [4.1–15.2] | 2,808 | 14,916 | 19 [11.9–27.1] | 17,441 | 14,313 | 28 [19.6–37.0] | 34,607 | 12,815 |
| 1 | 50% | 21 [13.6–29.3] | 3,470 | 27,760 | 32 [23.2–41.3] | 17,893 | 26,104 | 39 [29.7–48.6] | 36,326 | 22,918 |
| 3 | 25% | 51 [41.3–60.7] | 6,991 | 44,385 | 63 [53.5–72.1] | 41,538 | 42,092 | 83 [75.2–98.8] | 70,003 | 37,313 |
| 3 | 50% | 71 [61.9–79.5] | 8,094 | 81,583 | 90 [83.6–95.2] | 40,248 | 77,020 | 93 [87.4–97.2] | 73,553 | 66,109 |
| 5 | 25% | 79 [70.7–86.4] | 10,187 | 75,135 | 93 [87.4–97.2] | 59,811 | 70,873 | - | - | - |
| 5 | 50% | 98 [94.5–99.8] | 12,491 | 139,804 | 99 [96.2–100] | 58,203 | 129,115 | - | - | - |

offer long-term LF control. Control probability ranged from 51% [95% Credible Interval (CrI) 41.3–60.7] (25% HTA, 100m treatment radius) to 93% [95% CrI 87.4–97.2] (50% HTA, 1km treatment radius). The five-team strategies were the most effective household-based intervention. For 25% HTA, a 100m treatment radius offered a 79% [95% CrI 70.7–86.4] control probability, while the 500m treatment radius had a 93% [95% CrI 87.4.7–97.2] control probability. For the 50% HTA, both radii had an estimated control probability ≥ 98%.

The second household-based strategy, where only a single household member was tested, proved largely ineffective and unable to control LF. On average, testing a single household member resulted in five and a half times fewer tests than the regular testing strategy, however, this decrease in testing led to a large (approximately two thirds) decrease in control probability (S3 Fig).

## Probability of control and the number of tests and treatments

The various treatment strategies involved a trade-off between the number of treatments and tests. Importantly, MDA-only strategies were assumed to only involve treatment. This is a simplification as some community testing would be required in sentinel sites during MDA. If only the number of treatments are taken into account, the household-based strategies for both 25% and 50% HTA, offered a far greater control probability per treatment than MDA (Fig 4). This is exemplified by comparing the five-team strategy with the realistic MDA coverage. The five-team strategy with 50% HTA and 100m treatment radius required, on average, 12,491 treatments for a 98% [95% CrI 94.5–99.8] control probability, while four rounds of MDA with 73% coverage required, on average, 136,800 treatments for a comparable level of control.

Despite the minimal use of treatments, the cost of household-based strategies should account for testing. For all household-based strategies, doubling the proportion of households that were offered testing (50% vs 25% HTA) led to a significant increase in control probability and number of tests administered, while the number of treatments remained approximately

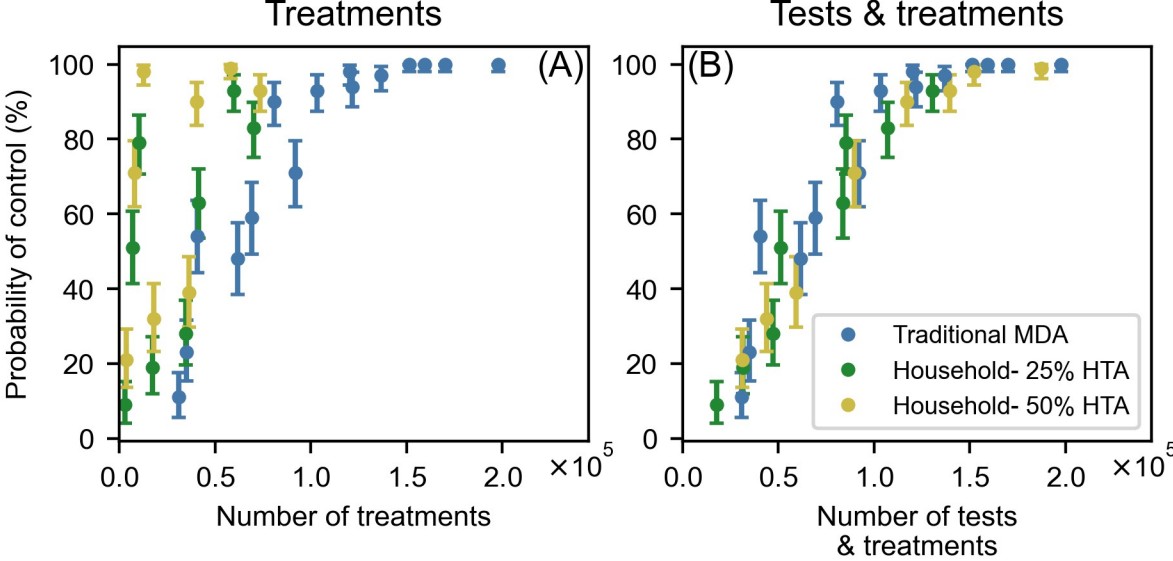

**Fig 4. LF control probability vs number treatments and tests for community-wide triple-drug MDA and household-based strategies.** Control probability was the proportion of simulations in which mf prevalence decreased from 2030 to 2035 and the household-based strategies offered testing to either 25% or 50% of households (HTA). (A) Control probability as a function of the average number of treatments per strategy. (B) Control probability as a function of the average number of tests and treatments per strategy.

constant. When testing is accounted for, MDA and the household-based strategies offered similar probabilities of control per treatment and test (Fig 4). Using the example from above, the five-team strategy with 50% HTA and 100m treatment radius, required, on average, 152,295 tests and treatments compared to the 136,800 treatments required, on average, by MDA with 73% coverage.

This trade-off between testing and treatment can also be seen in different household-based strategies. An increase in treatment radius led to a large increase in treatments per positive test, as more households were eligible to receive treatment. The small 100m treatment radius, on average, used five times fewer treatments than the 500m treatment radius did, but was only effective with the highest levels of testing. The increase in treatment radius also reduced the average number of tests administered, as the larger treatment radii meant more people were treated for each positive household and were therefore ineligible for testing for the next 180 days. This effect was most pronounced with the 1km treatment radius, that on average required 17.5% fewer tests than the same strategies with a 100m treatment radius.

### 1% antigen threshold

In a majority of simulated strategies, territory-wide antigen prevalence was estimated to fall below the 1% threshold before 2035; however, this drop below the critical threshold did not necessarily ensure long-term LF control. Many strategies predicted a LF resurgence and had poor control probabilities despite this initial drop below the critical threshold. Of the 18 3D-MDA scenarios tested, all had an estimated 100% probability of reducing territory-wide antigen prevalence to below 1% before 2035. However, only half had an estimated control probability $\geq$ 90%. Similarly, all sixteen household-based strategies had an estimated $\geq$ 99% probability of reducing territory-wide antigen prevalence below 1% by 2035. Nevertheless, only ten of the sixteen strategies had an estimated $\geq$ 50% probability of long-term control, while only five strategies had an estimated $\geq$ 90% probability of control. The poor predictive value of the 1% target threshold for control probability was most pronounced with the workplace- and school-based interventions. Each of the seven tested strategies had an estimated $\geq$ 99% probability of territory-wide antigen prevalence falling below 1%, however, six out of the seven strategies had an estimated control probability < 5%. Further details of the relationship of the 1% antigen threshold and the control probability are shown in S1 Table.

### Discussion

Our study used an innovative spatially-explicit agent-based model to simulate MDA and various targeted LF surveillance and treatment frameworks in American Samoa to determine the most efficient pathways to long-term LF control. Without further intervention, we predict a high probability of LF resurgence, highlighting the need for further action to ensure LF elimination. School- and workplace-based approaches to surveillance and treatment were unlikely to provide the necessary intensity of intervention to adequately control LF in the long term. While these strategies were able to suppress transmission, prevalence rapidly increased upon cessation.

We found that triple-drug MDA offered a realistic route to LF control in American Samoa. All three simulated coverage levels of 65%, 73%, and 85%, offered an above 90% probability of long-term control; however, we found that MDA coverage had a significant impact on the requisite duration and efficiency of intervention. This phenomenon has been noted in many previous studies [9, 45–48]. MDA is not, however, without limitations. MDA is treatment intensive, as it requires a majority of the eligible population to be treated annually for years and, importantly, previous MDA efforts in American Samoa have been hampered by poor

coverage [22]. Therefore, if triple-drug MDA is continued in American Samoa, measures should be taken to improve coverage. If increasing MDA coverage and maintaining said higher coverage across multiple future MDA rounds is not feasible, there are alternative strategies that may have an improved chance of success.

We modelled a targeted household-based strategy, which (unlike MDA) is dependent on mass testing rather than treatment. For comparable control probabilities, the household-based strategy uses a fraction of the treatments that are required by MDA. With the household-based strategy, people may be more inclined to accept treatment, as treatment is only offered to antigen positive individuals and those who live within a certain distance of a positive case. This may make the LF threat more tangible and thereby mitigate some people's perceptions that they are not at risk of LF.

Unlike MDA, household-based strategies require large numbers of tests to efficiently treat the population in a low LF prevalence setting such as American Samoa. As such, only approaches with the most intensive testing and treatment, offered ≥ 90% control probability after five years of implementation. Despite the necessity of testing, a 50% HTA combined with the need to test every member of a selected household is potentially unrealistic, especially in larger villages like Pago Pago. Furthermore, five concurrent teams testing and treating the population for five years would be a significant undertaking for the small geographic area and population size of American Samoa and potentially too large a burden for the health system. Therefore, to accurately access the household-based strategy's effectiveness, the demands of testing should be considered alongside the strategy's highly efficient use of treatments.

Both MDA and the household-based strategy use similar numbers of tests and treatments for comparable control probabilities; however, the economic cost of the two is likely different as MDA is reliant on treatment while the household-based strategy is reliant on testing. In American Samoa, the drugs for MDA are donated. Although there are significant costs associated with distribution and surveys for MDA, the household-based strategy (with its high volume of testing alongside its lengthy required duration of intervention) is likely more costly than traditional MDA. For MDA, the economic savings are greatest with scaled up interventions [49]; therefore, of the modelled MDA strategies, the strategies with the highest coverage have a double benefit of reduced administrative costs alongside the population having to participate in fewer rounds of MDA.

The successful MDA and household-based strategies relied on high levels of coverage. The requirement of high coverage coupled with the strategies' several year time spans, indicates that both strategies should include a locally engaged contingent, as studies have found that, both locally in American Samoa [22] and neighbouring Samoa [50], and globally [51–57], community engagement and participation is essential for program success. In American Samoa, there is existing framework (INdigenous Samoan Partnership to Initiate Research Excellence (INSPIRE)) that weaves Indigenous and Western knowledge, which has developed strategies to increase screening rates for colorectal cancer [58]. A similar approach could be taken with LF. In the Solomon Islands, a model for LF control in a post-elimination setting was developed and highlighted the utility of engaging with local health professionals [59]. Similarly, in Zambia, researchers found that an LF framework that included a locally engaged element, with community members such as local health professionals and traditional leaders, increased the trust and uptake in morbidity management services [60]. Therefore, local engagement is essential to increase community participation and thereby ensure a strategy's success.

In American Samoa, where *Aedes* is the primary vector, the WHO has set a target threshold of 1% antigen prevalence. In this study, we did not observe this threshold behaviour; instead we found that a territory-wide antigen prevalence of below 1% was a poor indicator of an

intervention's long-term success. This result has been observed in other modelling studies, which have found transmission dynamics at low prevalence and population size hard to predict [61, 62] and that singular prevalence thresholds are often insufficient for predicting resurgence [62]. There are possible explanations for this behaviour. Firstly, the lack of critical threshold behaviour in American Samoa could be caused by the local dynamics of the highly efficient *Aedes* vector, as thresholds are influenced by vector species, abundance and biting rate [8, 63]. Secondly, it is unclear what the suitable spatial scale for the threshold should be in a setting where residual infections are spatially heterogeneous. If the scale is territory-wide, the threshold will likely be met; however, if we consider prevalence at the village or neighbourhood scale where the population may be $< 100$, a single case will be sufficient to exceed the threshold. As GEOFIL includes geographic information on household, workplace and school locations, as well as daily commuting networks, the question of the appropriate spatial scale is especially pertinent.

Further modelling using GEOFIL could be undertaken to investigate other targeted testing and treatment options that may offer a high control probability while requiring less intensive testing than the household-based strategy. For example, MDA could be combined with the household-based strategy. American Samoa could undergo several further rounds of triple-drug MDA, followed by the post-MDA assessment surveys. These surveys could identify villages with continued transmission and the household-based strategy could then be implemented with a focus on these higher prevalence villages, thereby offering treatment and providing continued monitoring of these hot-spots. Similarly, annual workplace testing and treatment could be combined with MDA, to locate ongoing transmission after MDA and treat positive cases. Instead of solely relying on human testing to assess LF prevalence, molecular xenomonitoring (MX), which involves monitoring of filarial DNA in trapped mosquitoes, could be implemented to reduce the human testing burden [64]. MX has been shown to be a promising tool in deciding when to stop MDA and in conducting post-MDA surveys in American Samoa [18]. If used in conjunction with targeted surveillance and treatment approaches, the economic and social burdens of testing may be greatly diminished. Furthermore, GEOFIL could be expanded to include geographic factors, such as urban- and tree-cover, as determinants of transmission to potentially increase the heterogeneity of transmission events. This could produce more realistic predictions transmission and allow for a more accurate investigation of the effects of vector control based interventions.

There are limitations to the modelling presented in this study. Due to the limited number of recent community surveys in American Samoa, GEOFIL was initialised with prevalence from a 2010 community survey and was only fitted using antigen and mf data from two other community surveys (2014 and 2016 [17, 19]). Prevalence was low during these surveys, due to the 2000–2006 annual MDA rounds, thus the model has not been validated against survey data when prevalence was high. Therefore, the model may not correctly predict LF dynamics at higher prevalence; however, the model should correctly model elimination scenarios when prevalence is low. Furthermore, GEOFIL would benefit from fitting to post-MDA data, such as the community surveys after the recent 3D-MDA rounds in American Samoa. There are also limitations to GEOFIL's ability to reproduce the differences in prevalence between females and males. Previous community surveys in American Samoa have found antigen prevalence in males significantly higher than that in females [17], however, the gender antigen prevalence difference is not as pronounced in GEOFIL. This will cause an underestimation of the effectiveness of workplace-based strategies, as males have a higher labour force participation rate [39]. GEOFIL also assumes no systematic non-treatment or non-testing in any individuals or households. The lack of systematic non-treatment and non-testing may have caused an overestimation in an intervention's effectiveness [65].

## Conclusion

American Samoa has so far not interrupted LF transmission despite multiple recent rounds of MDA [20]. To prevent future LF resurgence, American Samoa requires further intervention through either traditional MDA or more novel targeted surveillance and treatment strategies. While household-based targeted strategies require fewer treatments than MDA, for comparable levels of LF control, they are far more testing intensive and may prove an economic burden. Therefore, the optimal policy needs to be investigated, to determine the best balance of long-term LF control with potential programmatic cost. Additionally, future modelling should be undertaken to investigate combining MDA with targeted interventions, to take advantage of each strategy's strengths to ensure the most efficient pathway to long-term LF elimination in American Samoa.

## Supporting information

**S1 Text. GEOFIL synthetic population model.** Description of the synthetic population model of American Samoa.
(PDF)

**S2 Text. GEOFIL transmission dynamics.** Description and underlying equations of the lymphatic filariasis transmission model.
(PDF)

**S3 Text. GEOFIL initialisation.** Description of the model initialisation process.
(PDF)

**S4 Text. GEOFIL fitting.** Description of the approximate Bayesian computation fitting procedure used in the model.
(PDF)

**S1 Fig. Results of the original ABC fitting.** Priors and posterior distributions for the original three ABC-fitted parameters. Both the priors and posteriors are kernel-smoothed densities. (A) The mean number of *bites* received per person during working hours ($b_d$). (B) The probability an infective bite would transmit one third-stage larva that will survive to maturity ($p_1$). (C) The probability an infective bite would transmit two third-stage larvae with one of each sex that will survive to maturity ($p_2$).
(TIF)

**S2 Fig. Results of ABC fitting—transformed parameters.** Prior and posterior distributions for the three transformed parameters. The priors and posteriors are kernel-smoothed densities. (A) The total daily transmission rate ($\beta_t$). (B) The ratio of the working hour transmission rate to the total daily transmission rate ($\beta_d/\beta_t$). (C) The probability a transmission event will transmit one L3 larva ($p$).
(TIF)

**S3 Fig. Household-based strategy with single household member testing.** Comparison of the two household-based strategies where every household member was tested, and the modified household-based strategy where only a single member of a household was tested. As the testing of a single person was likely to find fewer mf or antigen positive persons than testing the entire household, we only modelled the single person testing with the most effective whole household testing strategies. (A) Comparison of the effectiveness of the three-team strategies. (B) Comparison of the effectiveness of the five-team strategies.
(TIF)

**S1 Table. 1% antigen threshold as a predictor of intervention's success.** Probability of territory-wide antigen prevalence falling below 1% at any time and control probability, for MDA, household-based strategies, and school- and workplace-based strategies. The uncertainties for the control probabilities and antigen <1% are quantified with a 95% credible interval.
(PDF)

**S2 Table. Assumptions used in GEOFIL on the effectiveness of both 3D-MDA/treatment and 2D-MDA/Treatment for all strategies.** 3D-MDA/treatment (IDA) values are based upon the assumptions from Irvine et al [66].
(PDF)

**S3 Table. Model parameters for transmission dynamics.**
(PDF)

## Acknowledgments

We would like to thank Dr Zhijing Xu, for developing the original version of GEOFIL. We would also like to thank all field team members and everyone at the American Samoa Department of Health who provided assistance with the field studies in 2010, 2014, and 2016, that were used to fit GEOFIL.

## Author Contributions

**Conceptualization:** Callum Shaw, Angus McLure, Patricia M. Graves, Colleen L. Lau, Kathryn Glass.

**Formal analysis:** Callum Shaw, Angus McLure, Kathryn Glass.

**Methodology:** Callum Shaw, Angus McLure, Kathryn Glass.

**Software:** Callum Shaw, Angus McLure.

**Supervision:** Angus McLure, Kathryn Glass.

**Visualization:** Callum Shaw.

**Writing – original draft:** Callum Shaw.

**Writing – review & editing:** Callum Shaw, Angus McLure, Patricia M. Graves, Colleen L. Lau, Kathryn Glass.

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
