## [Decision Letter · Decision Letter 0]

17 Oct 2022

Dear Mr Shaw,

Thank you very much for submitting your manuscript "Lymphatic filariasis endgame strategies: Using GEOFIL to model mass drug administration and targeted surveillance and treatment strategies in American Samoa" for consideration at PLOS Neglected Tropical Diseases. As with all papers reviewed by the journal, your manuscript was reviewed by members of the editorial board and, in this case, by two independent reviewers. In light of the reviews (below this email), we would like to invite the resubmission of a significantly-revised version that takes into account the reviewers' comments. 

We cannot make any decision about publication until we have seen the revised manuscript and your response to the reviewers' comments. Your revised manuscript is also likely to be sent to reviewers for further evaluation.

Sincerely,

Prof. María-Gloria Basáñez, PhD, MSc

Academic Editor

Cinzia Cantacessi

Section Editor

Reviewer's Responses to Questions

**Key Review Criteria Required for Acceptance?**

**Methods**

-Are the objectives of the study clearly articulated with a clear testable hypothesis stated?

-Is the study design appropriate to address the stated objectives?

-Is the population clearly described and appropriate for the hypothesis being tested?

-Is the sample size sufficient to ensure adequate power to address the hypothesis being tested?

-Were correct statistical analysis used to support conclusions?

-Are there concerns about ethical or regulatory requirements being met?

Reviewer #1: (No Response)

Reviewer #2: The study appears to be well designed; however, it is impossible to judge the validity of the results because the transmission model is not described, and is also poorly described in the reference given for the model's origin. The key features which need to be described are the mechanism for sexual reproduction (this is vital as the paper discusses critical threshold behaviour which is only present if this mechanism is present); a form of negative density dependence (to prevent the mean burden in the population growing unbounded); if there is heterogeneity in exposure to biting (to create over-dispersed distributions of burdens). These could be addressed by summarising each step of the modelled life cycle of the parasite. The paper also needs more detail on how the model is fitted to data, in particular the precise form of the summary statistic used in the ABC procedure, and which form of ABC is being used (i.e. is the tolerance fixed or has an adaptive method been used?). It also isn't clear whether simulations are run using samples from the posterior distribution of parameters or from point estimates, or if the model is assumed to be stationary at the beginning of the simulations in 2010. I also have concerns about the low number of replicates used to generate results, which appears to be 100. Given this low number, the results should be presented with an estimate of uncertainty (e.g. the control probabilities in Table 3).

**Results**

-Does the analysis presented match the analysis plan?

-Are the results clearly and completely presented?

-Are the figures (Tables, Images) of sufficient quality for clarity?

Reviewer #1: (No Response)

Reviewer #2: Tables and figures are well presented. Given the spatial aspect of the study, it might be a nice addition to see maps of prevalence (if possible).

**Conclusions**

-Are the conclusions supported by the data presented?

-Are the limitations of analysis clearly described?

-Do the authors discuss how these data can be helpful to advance our understanding of the topic under study?

-Is public health relevance addressed?

Reviewer #1: (No Response)

Reviewer #2: It is not possible to judge if conclusions are supported.

**Editorial and Data Presentation Modifications?**

Reviewer #1: (No Response)

Reviewer #2: (No Response)

**Summary and General Comments**

Reviewer #1: This manuscript uses a spatially-explicit agent-based model which is specific to American Samoa, to find the optimal treatment strategy for LF. The findings are important for the 2030 elimination target in low prevalence settings. However, the ‘’Methods’’ section needs a major revision, as the model, fitting method, and prevalence data are not clearly described. The authors have included ‘’ probability of control and the number of tests and treatments’’ (line 257) and state that the results are summarized in Figure 4. However, I couldn’t find Figure 4 in the manuscript.

1. In the Abstract and throughout the manuscript, I would suggest using IDA for the triple drug as opposed to 3D-MDA. Similar comment for 2D-MDA which can be IA or DA. By using 2D-MDA it is not clear which treatment was used (IA or DA?).

2. In the Abstract, clarify whether the coverage used is for the whole population or for the eligible population.

3. In the Abstract, regarding the statement: ’All interventions were finished…’’, the author should clarify that these are simulated interventions? 

4. Line 49: Is this elimination of transmission (EoT) or elimination as a public health problem (EPHP)?

5. Line 78: Regarding the statement: ‘’However, alternative strategies should…’’, this is a statement that needs additional explanation. Which are these alternative strategies and what is their impact on the elimination target?

6. Line 97: Briefly describe the model.

7. Line 100: Give more information on the data used (i.e, demography, prevalence, etc). Were these data collected 4 years after the last MDA? 

8. Line103-108: Give more details on the fitting method.

9. Line 185: Why 100 runs?

10. Line 201: Regarding Figure S1: Fix the x-axis, 1e-3 looks odd at the end. Add more values on the y-axis (for the density).

11. Line 2016: Regarding Figure 3: Add a horizontal line at 1% threshold. Also, add more values on the x-axis. Currently, the x-axis has the years 2020 and 2030 only. 

12. Line 263: I cannot find Figure 4 in the manuscript.

13. General comment about the figures: Please follow the journal’s guideline when formatting them.

Reviewer #2: The paper is a novel and ambitious contribution to an import policy question (in particular the spatial aspect of transmission is a very nice feature), but I cannot judge if the conclusions are fully supported by the results of their simulations. The paper has potential if the methods were to be adequately described.

PLOS authors have the option to publish the peer review history of their article (what does this mean?). If published, this will include your full peer review and any attached files.

Reviewer #1: No

Reviewer #2: No
---

## [Decision Letter · Decision Letter 1]

22 Feb 2023

Dear Mr Shaw,

Thank you very much for submitting your revised manuscript "Lymphatic filariasis endgame strategies: Using GEOFIL to model mass drug administration and targeted surveillance and treatment strategies in American Samoa" for consideration at PLOS Neglected Tropical Diseases. As with all papers reviewed by the journal, your manuscript was reviewed by members of the editorial board and by several independent reviewers. The reviewers appreciated the attention to an important topic. Based on the reviews, we are likely to accept this manuscript for publication, providing that you modify the manuscript according to the review recommendations. 

Many thanks for implementing the changes suggested by the referees and revising your MS. The second reviewer has raised a small number of outstanding issues that need to be addressed. Please see attached review responding to your revisions, and highlighting some of the issues that still need to be addressed.

Sincerely,

Prof. María-Gloria Basáñez, PhD, MSc

Academic Editor

Prof. Cinzia Cantacessi

Section Editor

Reviewer's Responses to Questions

**Key Review Criteria Required for Acceptance?**

**Methods**

-Are the objectives of the study clearly articulated with a clear testable hypothesis stated?

-Is the study design appropriate to address the stated objectives?

-Is the population clearly described and appropriate for the hypothesis being tested?

-Is the sample size sufficient to ensure adequate power to address the hypothesis being tested?

-Were correct statistical analysis used to support conclusions?

-Are there concerns about ethical or regulatory requirements being met?

Reviewer #1: (No Response)

Reviewer #2: see attached

**Results**

-Does the analysis presented match the analysis plan?

-Are the results clearly and completely presented?

-Are the figures (Tables, Images) of sufficient quality for clarity?

Reviewer #1: (No Response)

Reviewer #2: see attached

**Conclusions**

-Are the conclusions supported by the data presented?

-Are the limitations of analysis clearly described?

-Do the authors discuss how these data can be helpful to advance our understanding of the topic under study?

-Is public health relevance addressed?

Reviewer #1: (No Response)

Reviewer #2: see attached

**Editorial and Data Presentation Modifications?**

Reviewer #1: (No Response)

Reviewer #2: (No Response)

**Summary and General Comments**

Reviewer #1: All my comments have been addressed with acceptable responses.

Reviewer #2: The revised paper has addressed most of my concerns, and is much improved. 

I have a few minor comments which still need to be addressed in the attached doc.

PLOS authors have the option to publish the peer review history of their article (what does this mean?). If published, this will include your full peer review and any attached files.

Reviewer #1: No

Reviewer #2: No

Figure Files:

Data Requirements:

Reproducibility:

References

---

## [Editor Report · Decision Letter 2]

29 Apr 2023

Dear Mr Shaw,

We are pleased to inform you that your manuscript 'Lymphatic filariasis endgame strategies: Using GEOFIL to model mass drug administration and targeted surveillance and treatment strategies in American Samoa' has been provisionally accepted for publication in PLOS Neglected Tropical Diseases.

Best regards,

Prof María-Gloria Basáñez, PhD, MSc

Academic Editor

Prof Cinzia Cantacessi

Section Editor

---

## [Editor Report · Acceptance letter]

12 May 2023

Dear Mr Shaw,

We are delighted to inform you that your manuscript, "Lymphatic filariasis endgame strategies: Using GEOFIL to model mass drug administration and targeted surveillance and treatment strategies in American Samoa," has been formally accepted for publication in PLOS Neglected Tropical Diseases.

Best regards,

Shaden Kamhawi

co-Editor-in-Chief

Paul Brindley

co-Editor-in-Chief
